# Exploring the Sequential-Selective Supercritical Fluid Extraction (S^3^FE) of Flavonoids and Esterified Triterpenoids from *Calendula officinalis* L. Flowers

**DOI:** 10.3390/molecules28207060

**Published:** 2023-10-12

**Authors:** Sirine Atwi-Ghaddar, Lydie Zerwette, Emilie Destandau, Eric Lesellier

**Affiliations:** Institute of Organic and Analytical Chemistry (ICOA), University of Orléans, CNRS UMR 7311, 45100 Orléans, France; sirine.atwi-ghaddar@univ-orleans.fr (S.A.-G.); lydie.zerwette@univ-orleans.fr (L.Z.); emilie.destandau@univ-orleans.fr (E.D.)

**Keywords:** extraction scale-up, marigold, narcissin, Box-Behnken design

## Abstract

One of the many advantages of supercritical fluid extraction (SFE) is the possibility of using it in sequential and selective approaches. This is due to the use of a dynamic extraction mode in addition to the possibility of altering the composition of the modifier during the extraction process. In this study, the optimization of *Calendula officinalis* L. extraction of non-polar and polar compounds was achieved using three-level Box-Behnken designs (BBD). For non-polar compounds, the factors were pressure, temperature, and EtOH percentage. As for the polar compounds, the three variables were temperature, the total modifier percentage, and H_2_O added in the modifier as an additive. The recovery of selectively rich extracts in triterpendiol esters and narcissin was possible using a sequential two-step SFE. The first step was performed at 80 °C and 15% EtOH, and the second at 40 °C and 30% EtOH:H_2_O 80:20 *v:v* with a total of 60 min of extraction. Additionally, the SFE extraction of non-polar compounds was scaled up on a pilot-scale extractor, demonstrating similar results. Finally, the SFE results were compared to ultrasound-assisted extraction (UAE).

## 1. Introduction

Plant-derived extracts have long provided a reliable source to supply pharmaceutical, nutraceutical, and cosmetic products that are rich in bioactive compounds. After the identification of the plant and compounds of interest, a suitable extraction procedure allowing the best final extract activity and yield should be selected.

Therefore, numerous traditional and modern extraction methods have been developed and used in several studies in this regard. Soxhlet, maceration, or hydro-distillation are among the techniques that are regarded as traditional, the advantages of which methods are the simplicity of use and low cost. However, these methods have some limitations, such as the necessity for large volumes of solvent and extended extraction times that might, in some cases, result in a low yield of recovery. In addition, the selective extraction of bioactive chemicals can be achieved using several solvents in series [1,2]. In this regard, green and modern extraction methods have emerged. Primarily, this was on an analytical scale that later scaled up on a pilot and industrial level. This allowed the use of less solvent, faster extraction times, and, often, a higher extract of purity [3,4,5,6].

One of these green extraction techniques is the use of supercritical carbon dioxide (SC-CO_2_) for the extraction of plant compounds; its low polarity made possible the replacement of toxic solvents like hexane while extracting non-polar compounds like triglycerides, carotenoids, fatty acids, and essential oils [7,8,9,10]. Moreover, the use of polar modifiers like ethanol (EtOH) and EtOH/water mixtures modulate its polarity in order to access more polar molecules like polyphenols [11,12,13,14]. This flexibility of SC-CO_2_’s polarity allowed the extraction of a larger spectrum of chemicals with different polarity and consequently different bioactivities and skin care properties while using the same plant biomass with decreasing sample treatment steps.

*Calendula officinalis* L. (Asteraceae), also known as pot marigold, is used as a traditional therapeutic plant for the treatment of various diseases [15,16,17,18,19,20,21,22,23]; this is due to its abundance of secondary metabolites. The phytochemical composition of marigold flowers has been extensively studied for both polar and non-polar compounds.

The composition of marigold’s lipophilic extract has been known since the late 1960s [24]. Active terpenoids such as ψ-taraxasteol, lupeol, erythrodiol, calenduloside, cornulacic acid acetate, faradiol, calendiladiol, maniladiol, β-amyrine, and arnidiol have been frequently emphasized for their abundance in the plant [24,25,26,27,28,29]. Most of these triterpenoids are found in an esterified form. In relation to this, a quantitative study of 10 varieties of *Calendula officinalis* L. dichloromethane extracts studied the content of triterpendiol monoesters; it identified faradiol-3-*O*-palmitate, faradiol-3-*O*-myristate, faradiol-3-*O*-laurate as major compounds. In addition, arnidiol-3-*O*-palmitate, arnidiol-3-*O*-myristate, arnidiol-3-*O*-laurate, calenduladiol-3-*O*-palmitate, calenduladiol-3-*O*-myristate and calenduladiol-3-*O*-laurate were also identified as minor ones [25].

Marigold’s yellow-to-orange color can be attributed to its carotenoid content [30]. Pigments like neoxanthin, luteoxanthin, antheraxanthin, flavoxanthin, mutatoxanthin, lactucaxanthin, lutein, zeaxanthin, rubixanthin, lycopene, γ-carotene, α-carotene, and β-carotene have been identified in the flower. However, the composition of carotenoids may differ depending on the variety and color of the flower [31]. The analysis of carotenoids found in the petals of six different cultivars identified 19 carotenoids, with 10 being unique to orange-colored cultivars. However, it was recurrent to see that luteoxanthin, flavoxanthin, and, in some cases, lutein were the most abundant [30].

The lipid content in *Calendula officinalis* L. seed extracts has been previously examined [32,33,34,35,36]. The fatty acid content in the petals of flowers like lauric, myristic, palmitic, stearic, oleic, linoleic, and linolenic acids has also been reported. It was also noted that the major fatty acids were the palmitic and myristic ones [37]. In addition, it was found that the petals contained tocopherol in α, β, and γ forms [38,39].

Many extraction methods have been applied to extract lipophilic compounds from marigold flowers. Some noted studies used Soxhlet with diethyl ether [40], and others in UAE with dichloromethane [41]. However, the application of supercritical fluids has been more frequent over the years; this is due to non-polar compounds’ compatibility with SC-CO_2_ polarity [42,43,44,45,46,47,48,49]. The extraction kinetics model of lipophilic compounds (i.e., oleoresin) was studied; it was determined that the increase in pressure improved the extraction yield, while an increase in the solvent flow rate decreased it. The temperature influence showed an increase in the yield only when the pressure was higher than 15 MPa [43]. In addition, major anti-inflammatory triterpendiol esters of marigold flowers (faradiol 3-*O*-laurate, palmitate, and myristate) were extracted using SFE with a pressure of 50 MPa and a temperature of 50 °C for 3 h; this resulted in an extraction yield of 5% for the dried extract with a recovery of 85% for the total faradiol esters from the crude herb [44]. Lastly, a theoretical model of the scale-up extraction of these molecules using SC-CO_2_ extraction was obtained, and it determined that a good simulation of kinetic behavior should be produced while maintaining a constant CO_2_ residence duration in vessels of various size [46].

For polar compounds, flavonoids and polyphenols have been reported in marigold extracts like calendoflavoside, isoquercitrin, rutin, isorhamnetin, quercetin, narcissin (isorhamnetin-*3-O*-rutinoside), astragalin, hesperidin and kaempferol [27,50,51,52]. Finally, flavonol glycosides like isorhamnetin 3-*O-β-D*-glucopyranoside, quercetin 3-*O-β*-D-glucopyranoside, and quercetin *3-O-β-D*-galactopyranoside [53] were identified. This is in addition to some coumarins like scopoletin, umbelliferone, esculetin [54].

It was demonstrated that the yield of polyphenols is associated with the antioxidant bioactivity of the extracts [55,56]. For the polar compounds of marigolds, many studies have been conducted using various extraction techniques. Microwave-assisted extraction (MAE) was optimized using a response surface methodology (RSM) with three factors: temperature, extraction time, and a solvent/solid ratio. The extracts were evaluated, and it was reported that the solvent (EtOH) concentration had the highest impact on the total phenolic content (TPC) and antioxidant activities [55]. Another study tested several extraction methods for polar compounds, including homogenizer-assisted extraction (HAE), maceration (MAC), Soxhlet, and ultrasound-assisted extraction (UAE). These methods all used methanol (MeOH) as an extraction solvent. It stated that the flower extracts provided the highest total phenolic content. The yields were examined for the four extraction techniques, and the study determined that the phytochemical profile of the extracts was different for each extraction process [52].

The sequential and selective extraction of non-polar then polar compounds from *Calendula officinalis* L. has been explored before, mainly due to the different properties found in each fraction. This is because the hydrophilic fraction offers antioxidant activity, and the lipophilic one offers anti-edematous and anti-inflammatory properties [41].

The sequential maceration at room temperature for two days with n-hexane, dichloromethane, acetone, ethyl acetate, methanol, and distilled water allowed different polarity extracts to be recovered for the flowers [57]. In addition, two different extraction processes of SC-CO_2_ extraction followed by UAE with EtOH:H_2_O 50:50 was reported as a green extraction method [42].

However, these approaches required either long extraction times, the drying of the extracts for the next step, and changing the solvents to fit the polarity of the compounds or using two different extraction processes. This is a valid approach for characterizing and testing the activities of the fractions, though an industrial extraction for cosmetic or pharmaceutical use requires a reduction in the process step and in certain cases, the use of only green solvents.

The sequential-selective supercritical fluid extraction (S^3^FE) of different polarity compounds has allowed some of the drawbacks previously reported to be overcome [58,59]. In this paper, this approach was applied to *Calendula officinalis* L. flowers with the objective of limiting the extraction solvents’ volume and unit operations with the use of the dynamic extraction mode. A rationalized approach is described in this paper. Both steps were optimized using the experimental design model adapted for the polarity of the targeted compounds.

Finally, the extraction of non-polar compounds was scaled up from an analytical scale (1 g of plant) to a pilot sale extractor (100 g of plant). To our knowledge, this topic has never been addressed by other publications.

## 2. Results and Discussion

SFE applied to *Calendula officinalis* L. flowers was optimized based on two Box-Behnken designs (BBD). The first model targeted non-polar compounds (CNPE); the temperature and pressure were chosen for their ability to change the SC-CO_2_ density, and the EtOH percentage was used as a modifier for its capacity to modify the polarity of the extraction phase (Table 1).

The preliminary screening of the EtOH percentage to be included in the CNPE design of the experiment (DoE) determined that ethanol was necessary for the extraction of triterpendiol esters from marigold in the pressure and temperature ranges chosen for this design. In addition, the second model targeted the extraction with polar compounds (CPE). Temperature, in terms of the total modifier percentage and water percentage added in the modifier, were selected as the three parameters to be investigated for the extraction of marigold flavonoids; this choice was based on previous work conducted for the extraction of caffeine and total catechins from green tea [11]. As for the pressure, in prior research, the extraction of two polar flavonoids was optimized with pressure serving as the optimization parameter with a range of 10 to 20 MPa; it was demonstrated that in that range, pressure was an insignificant variable for the extraction of such polar compounds [14]. This was mainly due to the fluid’s high density, especially with high percentages of modifiers (up to 30%). Therefore, the pressure was maintained constantly at 15 MPa throughout the present investigation and was not explored because it had little impact on the model’s response (compound yield).

### 2.1. Optimization of Calendula Non-Polar Extracts (CNPE)

#### 2.1.1. Statistical Analysis, Model, and Factor Significance

To achieve the selective high recovery of non-polar compounds without the co-extraction of polar ones, the optimization of triterpendiol esters extraction was carried out first. The total peak areas of major triterpendiol esters (faradiol myristate and faradiol palmitate) found in *Calendula officinalis* L. non-polar extracts (CNPE) were used as the response (*Y*).

The model’s determination coefficient (R^2^) was equal to 0.93, while the value of the adjusted determination coefficient (adj R^2^) was equal to 0.88. This signifies that the model was unable to account for 7% of the total variance. The R^2^ value demonstrated a highly strong correlation between the model’s expected and experimental response values and indicated that the theoretical model shows a good fit with experimental data. The relative standard deviation (RSD %) of the central point replicates (5 repetitions) for the peak area of both molecules and dried extract yield (mg/g) were equal to 12.42% and 4.42%, respectively. Furthermore, the Dixon statistical test was applied and showed no presence of outliers in responses, which was validated by the analysis of the residuals between predicted and experimental values. The results are shown in Table 1, and the correlation between the three variables (*X*_1_, *X*_2_, and *X*_3_) and the response (*Y*) is presented in Equation (1). The terms presented were chosen to obtain the highest pred R^2^; therefore, the predictive ability of the model was increased for the response. The quadratic terms and interaction between the temperature and modifier percentage were included, and the predicted R^2^ was equal to 0.54.
*Y* (mAU × min) = 7.976 × 10^7^ − 4.881 × 10^7^*X*_1_ + 3.73 × 10^8^*X*_2_ − 2.7664 × 10^8^*X*_3_ + 8.843 × 10^4^*X*_1_^2^ − 1.21 × 10^7^*X*_2_^2^ + 3.556 × 10^6^*X*_3_^2^ + 5.189 × 10^6^*X*_1_*X*_3_(1)

The results of the polynomial model’s analysis of variance (ANOVA) are summarized in Table 2, and the regression results indicate that this model is significant. Each factor’s significance was assessed using the probability associated with its F-value and *p*-value, and the percentage of each factor’s contribution to the model’s response was examined. The marigold regression model’s total sum for the faradiol ester peaks area had an F-value of 17.57 and a corresponding *p*-value of 0.001. A significant model is presented by both statistical terms.

The factors *X*_2_ (pressure) and *X*_2_^2^ (pressure × pressure) were significant, with a 15.3% and 14.7% contribution, respectively. This was also validated by their *p*-values of 0.0083 and 0.0092, respectively. However, the interaction term between the temperature and EtOH percentage (*X*_1_
*× X*_3_) had the highest significance, with a 41% contribution and a 0.0004 *p*-value. The separate term of temperature (*X*_1_) was non-significant to the model’s response with a 3.75% contribution and *p*-values of 0.1294; as for the EtOH percentage (*X*_3_), the factor was significant with a contribution of 11.8% and a *p*-value of 0.0160. This indicated that the response was influenced less by the distinct variations in temperature and EtOH percentage. However, the extraction yield of both the faradiol myristate and faradiol palmitate was significantly impacted by the interaction between the two variables. The interaction terms of *X*_1_
*× X*_2_ and *X*_2_
*× X*_3_ were excluded from the DoE, and their influence on the response was assessed during the analysis of the model. However, when they were included, these factors had a low significance and decreased the predicted R^2^ to 0.33).

#### 2.1.2. Effects of the Extraction Parameters on CNPE Assessed Using the Box-Behnken Design

Heat maps were used as a graphical representation of independent factors and dependent response variations; they were correlated to the response of marigold faradiol myristate and faradiol palmitate peak areas (mAU × min) for 30 min of the extraction time (Figure 1).

Figure 1a represents the effect of temperature (*X*_1_) and pressure (*X*_2_) at 10% of EtOH on the extraction yield of faradiol esters. The heat map demonstrated that the pressure’s influence on the extraction yield was weak and independent from the temperature, and the optimal pressure was equal to 15 MPa for the full temperature range studied. The same behavior was noticed with the pressure and interaction of the EtOH percentage.

Figure 1b illustrates the effect of temperature (*X*_1_) and the variation in the EtOH percentage (*X*_3_) on the response, indicating that both factors were influential in the extraction of faradiol esters. The increase in the EtOH percentage from 5 to 15% at 40 °C and 15 MPa had little effect on the peak areas of non-polar-targeted compounds, and their calculated value increased from 8.848 × 10^8^ to 9.056 × 10^8^ mAU × min (2.35%). Nonetheless, the same increase in EtOH at 80 °C resulted in an augmentation of peak areas from 3.945 × 10^8^ to 2.491 × 10^9^ mAU × min (531.43%). In addition, when comparing the results between 40 °C and 80 °C using 5% of EtOH, it showed that the temperature had the opposite effect, where a decrease of 55.41% was noticed in the yield with an increase in temperature. This could explain the influence of the interaction term *X*_1_
*× X*_3_ represented on the ANOVA analysis in Table 2, where the influence of high temperature is only related to the simultaneous increase in the EtOH percentage. In this case, the effect of density variation linked to temperature increase was disadvantageous for the yield increase at lower percentages of the modifier. However, at high percentages (15% EtOH), density variation was minimal, and the variation in solubility was predominant. Hence, extraction was favored at 80 °C. In conclusion, the combination of temperature and ethanol percentage had a highly significant influence on the results.

#### 2.1.3. Optimal Selective Extraction of Faradiol Esters

Based on Equation (1), the computed optimum parameters, offering a maximal yield of faradiol myristate and palmitate, were as follows: a temperature of 80 °C, a pressure of 15.52 MPa, and an ethanol percentage of 15%.

These conditions corresponded to the experiment n°8 (80 °C, 15 MPa, and 15% EtOH), for which the highest yield of faradiol palmitate and myristate was found with a peak area of 2.66 × 10^9^ mAU × min and a dry extract mass of 92.3 mg/g for biomass (Table 1). Other experiments that offered a close yield in the set were experiments n°2, 11, and 12. Additionally, Table 1 shows that these conditions (n°8) offer a selective extraction of non-polar compounds without co-extracting the polyphenols of marigold; the UC-DAD analysis of the experiment n°8 CNPE did not detect the extraction of polar compounds, especially the targeted molecule of narcissin. By contrast, the experiments n°2 (80 °C, 10 MPa, and 10% EtOH), n°7 (40 °C, 15 MPa, and 15% EtOH), n°11 (60 °C, 10 MPa, and 15% EtOH), and n°12 (60 °C, 20 MPa, and 15% EtOH) demonstrated a co-extraction of these polar molecules. The combination of all these values validated the relevance of experiment n°8 for the selective and efficient extraction of faradiol esters. One can note that the extracted amount of triterpenoid esters was significantly higher than the one reached using pure CO_2_ [26,60].

#### 2.1.4. Experimental Design Validation: Extraction Kinetics CNPE

The experimental design presented the results for 30 min of extraction time to validate these results while including the time factor in the response; extraction kinetics were investigated in several experimental conditions (Figure 2).

Figure 2a shows the extraction kinetics of faradiol esters for two EtOH percentages at 5 and 15%; both extractions were conducted at 15 MPa and 80 °C. The results validated that a high EtOH percentage is needed to attain a high extraction yield of faradiol esters for calendula flowers. Moreover, when comparing both yields at 30 min, 15% EtOH had a 175% higher percentage when compared to 5% EtOH. This validated the results of the experimental design, demonstrating that a high percentage of the modifier at 80 °C was needed to increase the yield of faradiol esters from marigold petals.

Figure 2b represents the extraction’s kinetics of faradiol esters at a constant temperature of 60 °C (median level), an EtOH of 15%, and three different pressure values: 10, 15, and 20 MPa. The extraction kinetics at the three pressure points had minimal variations for all extractions, with 15 MPa demonstrating the highest yield. This indicates that pressure modifications had a relatively slight effect on the yield of non-polar compounds extracted from marigold flowers at this modifier percentage.

In addition, the influence of temperature was demonstrated in Figure 2c. With a constant pressure and EtOH percentage of 15 MPa and 15%, the extraction kinetics of three different temperatures were investigated: 40, 60, and 80 °C. The results from the experimental design Figure 1a were validated as, with an 80 °C temperature, the highest extraction yield was obtained. In addition, at 40 and 60 °C, the yield kinetics had a similar extraction yield during the total extraction time. At 30 min, a yield increase of 79% was noted with the increase in temperature from 40 to 80 °C.

The kinetics behavior was similar to Figure 2b (at 60 °C), as within the first 30 min of extraction, 90% of faradiol esters were extracted compared to the total yield obtained at 60 min considered at 100%.

Finally, for all extraction kinetics, a comparable behavior of the yield’s percentage variation was observed. After the first 30 min, almost 90% of the yield was obtained, and for the second 30 min, only 10% was extracted. This might be explained by the matrix effect of the plant mass, where at the initial 30 min, the highest obtainable extraction yield within the conditions applied was reached, and any additional extraction time applied increased the extraction yield to only 10%. In most cases, the increase in extraction time in SFE allows for the convergence of the non-optimal yield to the yield of optimal conditions, meaning that theoretically, the highest extraction yield is reached faster with optimal conditions. Nonetheless, for less optimal conditions, the same extraction yield can be reached, but it requires longer extraction times and higher energy consumption [61]. However, for *Calendula officinalis* L., in the conditions range used in this study, the use of longer extraction duration for non-optimal conditions did not allow the same recovery yield to be reached compared to the one obtained for optimal ones (for instance, Figure 2a from 5 to 15% of modifier, Figure 2b or from 60 to 80 °C). Consequently, optimal conditions should be applied, and only 30 min is necessary to obtain almost 90% of the optimal extraction yield.

#### 2.1.5. Pilot-Scale Supercritical Extraction of CNPE

A pilot-scale extraction (PE) of 100 g of biomass was conducted to compare and validate the extraction results with the analytical-scale extractor (AE) of 1 g of biomass. The extraction conditions chosen for the investigation of scale-up were 60 °C, 15% EtOH, and 15 MPa. The temperature of 60 °C was chosen to favor the lower energy consumption required for the cosmetic industry and to facilitate the extraction of cell handling. A comparison between both extractors is presented in Figure 3.

The results in Figure 4 represent the cumulated fardiol myristate and faradiol palmitate yields normalized for 1 g of plant mass for both pilot and analytical scale extractors. When comparing the extraction kinetics of both AE and PE, the results in Figure 4a show that the analytical scale extractor offered a higher and faster extraction yield when compared to PE, with a 768% difference at around 15 min in time. However, the yield of PE slowly progressed to reach only a 6.42% difference with AE at 60 min.

However, this kinetics representation in relation to extraction time is not ideal when working with different extraction flow rates; thus, a comparison between both extractors was also conducted in function to the solvent-to-feed ratio F (mL/g). F was calculated following Equation (2):
F (mL/g) = [Flow rate (mL/min) × Extraction time (min)]/Plant mass (g)(2)

Indeed, when comparing AE and PE in function with the solvent-to-flow ratio allowed a higher understanding of the scale-up procedure. The results presented in Figure 4b show that both extractors had very similar results when flow rate, time, and plant mass were considered.

Finally, PE fractions were analyzed to determine if polar compounds, specifically narcissin, were extracted. Since none of these compounds were detected, it demonstrated that scale-up offered an effective and selective extraction of non-polar compounds.

#### 2.1.6. Comparison between Supercritical Fluid Extraction (SFE) and Ultrasound-Assisted Extraction (UAE) for CNPE

SFE was also compared with UAE for the extraction of non-polar compounds. UAE is regarded as an efficient and simple extraction technique. This comparison was conducted in equivalent conditions of organic solvent consumption: 13.5 mL of solvent volume equivalent to 15% of the modifier for 30 min at a 3 mL/min flow. Only using the modifier equivalent volume and not the whole CO_2_-EtOH volume was chosen to reduce the consumption of organic solvents. In the case of using the CO_2_-EtOH volume consumed for 30 min at 3 mL/min, 90 mL of organic solvent was used, which is a very high volume for only 1 g of plants and that later needed to be evaporated.

Based on the previous literature, three traditional extraction solvents were explored: heptane (hept), dichloromethane (DCM), and acetone (Ace). The results are presented in Figure 5.

For all extracts, faradiol myristate was the major faradiol ester. However, significantly different yields were obtained from each technique.

The comparison between SFE and UAE demonstrated that SFE had a significantly higher extraction yield compared to UAE regardless of the solvent. The extraction recovery of faradiol esters with UAE using Ace, DCM, and hept resulted in a relative yield percentage of 17, 23, and 10.4%, respectively, compared to the SFE yield considered at 100%.

This indicates that SFE is highly advantageous for the recovery of a high yield of non-polar compounds from *Calendula officinalis* L. and additionally requires the use of non-toxic solvents.

### 2.2. Optimization of Marigold Polar Compounds Extraction (CPE)

#### 2.2.1. Statistical Analysis, Model, and Factor Significance

Based on the second BBD, SFE, when applied to marigold flower polar compounds, was optimized. This temperature was selected because of its ability to impact the solubility of compounds in the SC-CO_2_, and the modifier and water percentages were chosen because of their capacity to increase the polarity of the extraction phase (Table 3). After the pilot-scale extraction of non-polar compounds (see Section 2.1.5), the plant mass was collected and used as treated biomass for the optimization of polar compounds extraction.

The model’s determination coefficient (R^2^) was equal to 0.97, while the value of the adjusted determination coefficient (adj R^2^) was 0.94. This signifies that the model was unable to account for almost 3% of the total variance. The R^2^ value demonstrates a highly strong correlation between the model’s expected and experimental response values and indicates that the empirical model showed a good fit with experimental data. The Dixon statistical test was applied to the data set and displayed the presence of one experiment as an outlier; this was validated by the analysis of the residuals between predicted and experimental values, which is a point represented by the repetition of the central point and was removed from the model. The relative standard deviation (RSD%) of the central point replicates (4 repetitions) for the concentration of narcissin (mg/g of plant), and after the elimination of one outlier was equal to 29.19%.

The results are shown in Table 3, with 16 experimental points left to create the model. The correlation between these three variables (*X*′_1_, *X*′_2_, and *X*′_3_) and their response *Y*′ (mg/g) is presented in Equation (3). The terms presented in this equation were chosen to obtain the highest possible predicted R^2^, therefore increasing the predictive ability of the model for its response. The quadratic terms and interaction between the modifier percentage and water percentage in the modifier were included, and the predicted R^2^ was equal to 0.72.
*Y*′ (mg/g) = 4.402 − 0.1134*X*′_1_ − 0.09875*X*′_2_ − 0.3241*X*′_3_ + 0.0009531*X*′_1_^2^ + 0.001737*X*′_2_^2^ + 0.01016*X*′_3_^2^ + 0.01313*X*′_2_*X*′_3_(3)

The results of the polynomial model’s analysis of variance (ANOVA) are summarized in Table 4, and the regression results indicate that the model is significant. Each factor’s significance was assessed using the probability associated with its F-value and *p*-value, and the percentage of each factor’s contribution to the model’s response was examined. The marigold regression model’s narcissin yield (mg/g) had an F-value of 34.3206 and a corresponding *p*-value of 0.0000. A highly significant model is presented by both statistical terms.

The factor *X*′_3_ (H_2_O percentage in modifier) and its quadratic term *X*′_3_^2^ (H_2_O% × H_2_O%) were significant, with 26.5% and 22.1% of the contribution, respectively. Their respective *p*-values of 0.0004 and 0.0008 validated this as well. However, the interaction term between the modifier’s percentage and water’s percentage (*X*′_2_
*× X*′_3_) displayed the highest significance and contributed 36.9% to the model’s response and presented a 0.0001 *p*-value. The temperature (*X*′_1_) had limited significance to the model’s response with a 3.02% contribution and *p*-values of 0.0887; as for the separate term of the modifier’s percentage (*X*′_2_), it was not significant with a contribution of 1.19% and a *p*-value of 0.2579.

#### 2.2.2. Extraction Parameters Influence on CPE Assessed with Box-Behnken Design

Heat maps were used as a graphical representation of the independent factors and dependent response interactions; they were correlated to the response of marigold flower polar compounds, such as narcissin (mg/g of plant), for 15 min of extraction time.

Since the temperature presented minimal influence on the response, Figure 6 presents the influence of the total modifier percentage and water percentage included as an additive in the modifier for narcissin yield. This demonstrates that both the water content and the modifier’s percentage had a favorable effect on the extraction yield of narcissin. When the temperature was set at 60 °C and no water was added to the modifier, narcissin yield was very low, 0.0 mg/g and 0.05 mg/g for the modifier percentage was equal to 10% (exp n°9), and 30% (exp n°10).

However, the increase in the modifier percentage at 20% H_2_O in the modifier increased the predicted yield from 0.42 mg/g to 5.07 mg/g (1107.14% increase). These results confirm the significant effect of water as an additive to extract polar compounds from marigold flowers, and the same conclusion was made for the total polar compounds. The high polarity of narcissin is most likely responsible for this behavior. Indeed, SC-CO_2_’s low polarity limits its ability to extract non-polar or moderately polar compounds. However, the extraction phase’s polarity was boosted by the addition of ethanol and water as modifiers, which increased the affinity of polar molecules like target polyphenols and raised the recovery yield.

#### 2.2.3. Optimal Selective Extraction of Narcissin

Based on Equation (3), the computed optimal extraction conditions to recover the maximum amount yield were the following: a temperature of 80 °C and 30% EtOH:H_2_O 80:20 *v:v*, which allowed for the recovery of 5.5 mg/g of narcissin and 14.9 mg/g of total polar compounds. These values were higher than the ones obtained using ethanol/water/propane diol mixtures [62]. However, since the temperature had a low significance to the model’s response, the yield at 40 °C and the same amount and composition of the modifier allowed for the recovery of 5.45 mg/g of narcissin and 14.8 mg/g of total polar compounds. This showed a minimal decrease in the yield, but it also allowed a significant reduction in energy consumption for the SFE of polar *Calendula officinalis* L. compounds.

In order to validate the experimental design results and determine the optimal extraction time, one hour of extraction kinetics was performed at 40, 60 and 80 °C with the use of 30% EtOH:H_2_O 80:20 *v:v* as a modifier (Figure 7). It shows that after 15 min of extraction time for a set temperature of 40, 60, and 80 °C, the narcissin yield was of 4.00, 3.08, and 3.80, respectively.

In addition, the cumulated yield at 30 min was 4.30, 4.28, and 4.33, respectively. The yield did not increase after this point, demonstrating that 30 min of extraction was sufficient for the extraction of polar compounds from *Calendula officinalis* L., validating that the use of 40 °C, which can recover similar amounts of narcissin with lower consumption of energy.

#### 2.2.4. Comparison between SFE and UAE for CPE

The UAE extraction of treated and non-treated biomass (the same batch of plant without a prior SFE of non-polar compounds) was conducted in an equivalent condition to the optimized SFE extraction of polar compounds from marigold, where 27 mL of EtOH:H_2_O solvents with 50:50 and 80:20 *v:v* were used, which is equivalent to a 30 min extraction with 30% of the modifier at a 3 mL/min flow rate. These results were compared with optimized SFE extraction conditions at 40 °C and 30% EtOH:H_2_O 80:20 *v:v* (Figure 8).

The comparison between the treated and non-treated biomass for UAE extraction demonstrated that prior treatment with the pilot-scale extraction of faradiol esters did not influence the extraction of narcissin or total polar compounds from *Calendula officinalis* L. flowers.

In addition, for the UAE of treated biomass, the increase in water percentage in the extraction solvent from 20 to 50% resulted in an increase in the extraction yield of total polar compounds and, specifically narcissin, by 50%, producing a yield that was comparable to that of SFE extraction. This increase was due to the polarity of the compounds, mainly narcissin, which is a glycosylated polyphenol with a log P of −1, making it a highly polar compound with a considerable affinity for extraction solvents with higher percentages of water.

However, extracts with a high water content (50%) are generally less stable microbiologically and require the evaporation of the extraction solvent and consumption of energy. Consequently, SFE, which only uses 6% of water in the extraction fluid, can offer an extract with a higher concentration of bioactive compounds while reducing the final water % in the extracts, subsequently favoring the stability of the extract and decreasing the need for further evaporation.

### 2.3. Validation of S^3^FE

The selective extraction of both non-polar compounds and polar compounds in *Calendula officinalis* L. was optimized using two separate BBDs. These results show that a high temperature of 80 °C combined with 15% EtOH allowed for the recovery of a high yield of faradiol esters (myristate and palmitate), and this residue could be extracted using 30% EtOH:H_2_O 80:20 *v:v* at 40 °C to recover the polar compounds.

To validate the sequential extraction of both steps, a sequential selective supercritical extraction (S^3^FE) was applied to marigold flowers in non-treated powder. The first step used 15% EtOH at 80 °C and the second step 30% EtOH:H_2_O 80:20 *v*:*v* at 40 °C; both steps lasted for 30 min, leading to a total extraction time of one hour, where fractions were collected every 15 min and a 5 min cool time was required between the two steps. All fractions were analyzed using SFC-ELSD, UC-DAD, and UHPLC-DAD to determine their content of molecules of interest. These results were illustrated by means of a percentage of yield for the total hour of extraction (Figure 9), and the results showed that a full selective and sequential extraction could be achieved with both steps.

The first step of 30 min ensured that a faradiol ester-rich extract was obtained within the first 15 min, and 87.4% of faradiol esters were extracted; the remaining amount was collected in the second 15 min fraction with 12.6%. The UC-DAD analysis showed no co-extraction of polar compounds, which validated the results of the experimental design. In the second step, the following 30 min allowed for the extraction of polar compounds, including narcissin. Similar extraction kinetics were noticed for the polar compounds. The first 15 min of the second step allowed for the extraction of 79.6% of polar compounds and 81.7% of the narcissin yield, which were both compared to the total yield recovered at the end of the extraction. The final 15 min of the second step allowed for the recovery of the remaining molecules of interest with a yield of 20.4% for polar compounds and 18.3% for narcissin; the yield of the polar compounds was equivalent to the amount of optimized and determined conditions of the BBD with 4.63 mg/g and 13.85 mg/g of narcissin and total polar compounds, respectively.

Consequently, the first step of CNPE did not significantly influence the recovery of polar compounds regarding the extraction kinetics in Figure 7, validating how the S^3^FE approach is applicable for the fast and sequential recovery of both non-polar and polar compounds from *Calendula officinalis* L. flowers.

## 3. Methodology

### 3.1. Plant Material

The dried flower of the marigold (*Calendula officinalis* L.) plant consisted of a light brown, yellow powder, which was provided by PMA 28 (Varize, France) and kept at room temperature in an airtight container.

### 3.2. Chemical and Reagents

Air Liquide (Fleury-les-Aubrais, France) provided CO_2_ gas. Acetonitrile (ACN), methanol (MeOH), dichloromethane (DCM), acetone (Ace), heptane (hept), dimethylsulfoxide (DMSO), and ethanol (EtOH) were provided by VWR (Fontenay-sous-Bois, France). Formic acid (FA) and methanesulfonic acid (MSA) were purchased from Sigma-Aldrich (Merck, Semoy, France). Ultra-pure water (H_2_O) was purified using a Milli-Q system from Sigma-Aldrich with a resistance of >18 MΩ.cm. The narcissin (Isorhamnetin-3-*O*-rutinoside) standard was used to identify and quantify the compound (Purity (HPLC) ≥99%) supplied by Extrasynthese (Genay, France).

### 3.3. SC-CO_2_ Extraction

For the analytical scale SFE, a Waters MV-10^®^ ASFE (Milford, MA, USA) was used for all extractions. In total, 1 g of the plant powder was combined with 1 g of diatomaceous earth powder from Sigma-Aldrich (Merck, Semoy, France) in a stainless-steel extraction vessel (5 mL). To filter the extract and completely fill the extraction cell, cotton was positioned at the top and bottom parts. A continuous dynamic extraction was used with a 3 mL/min flow rate. The amount of modifier added was calculated based on this overall extraction flow.

For the pilot-scale extraction of non-polar compounds, an SFE process (Tomblaine, France) extractor was used. This extractor was equipped with a CO_2_ and co-solvent pump. A gravity separator (S_1_) and a cyclone separator (S_2_) were used to collect the extracts. The extraction conditions were set to a 60 °C temperature, 15 MPa pressure, a total flow of 60 mL/min, and 15 % of EtOH (96% purity) added as a modifier to the SC-CO_2_. One liter of the stainless-steel extraction vessel was used and filled with 100 g of plant material combined with 100 g of diatomaceous earth. The total duration of the EtOH-modified extraction was 57 min, where 6 fractions were weighed and collected from S_1._ Approximately 50 g of the extract was collected for each fraction (around 9 min for each fraction). A final step (fraction 7) of 90 min with 100% SC-CO_2_ was applied to dry the plant mass, removing the residual EtOH and collecting extract residues. At the end of extraction, the content of the S_2_ for the total extraction duration was collected for analysis.

### 3.4. Ultra-Sound Assisted Extraction (UAE)

The UAE was performed in a Branson 3510 (Bransonic^®^ ultrasonic) bath from Sigma-Aldrich (Semoy, France), and the power of the extractor was equal to 130 W. Using 1 g of *Calendula offininalis* L. flower powder mixed with solvents in an equivalent volume consumption to the determined SFE optimized conditions, an extraction duration of 30 min was applied for all extractions.

To examine the extraction yields of calendula polar extracts (CPE), EtOH:H_2_O in an 80:20 and 50:50 *v:v* composition was utilized as an extraction solvent, with a volume of 27 mL. As for calendula non-polar extracts (CNPE), three non-polar solvents were used: heptane (hept), acetone (Ace), and dichloromethane (DCM), with a volume of 13.5 mL. All extractions were conducted in duplicate to evaluate the repeatability.

After sonication, the extracts were centrifuged at 10,000 rpm for 10 min at 25 °C and filtered using a 0.45 µm polyvinylidene difluoride (PVDF) syringe filter from Agilent Technologies (Les Ulis, France).

### 3.5. SC-CO_2_ Extracts Sample Treatment

To compare and determine the analytical-scale extract yields for the target compounds, all extracts were evaporated using a nitrogen flow for a maximum of 24 h. The non-polar extracts (CNPE) were diluted with 1 mL of a mixture of DCM:MeOH 1:1 *v:v* for SFC analysis. The polar extracts (CPE) were diluted in 12 mL of MeOH:DMSO:H_2_O 9:2:1 *v:v:v*, from which they were sonicated for 40 min for total solubilization, filtered using 0.45 µm PVDF and later diluted 4 times using H_2_O for UHPLC analysis.

For the pilot-scale extracts of non-polar compounds, all fractions in the volume were adjusted to 80 mL of EtOH to compare and determine the extraction yield. In total, 1 mL of each fraction was collected and evaporated using a nitrogen stream. The extracts were analyzed using the CNPE chromatographic method, and the results were normalized to 1 g of the extract to compare with the analytical-scale extractor.

### 3.6. Calendula Non-Polar Extracts (CNPE) Chromatographic Analysis

#### 3.6.1. Ultra-High Efficiency Low-Pressure/Supercritical Fluid Chromatography (UHLP/SFC-ELSD): Terpenoid Esters Analysis in CNPE

Shimadzu Corporation’s (Kyoto, Japan) Nexera UC system was used for the analysis of terpenoid esters. This system included an autosampler (SIL-30AC), column ovens (CT0-20AC), a photodiode array (PDA) detector (SPD-M20A), a back-pressure regulator (BPR) (SFC-30A), a carbon dioxide pump (LC-30ADsf), a modifier pump (LC-30AD), and an evaporative light scattering detector (ELSD-LT-III). Shimadzu Corporation, LabSolutions LCMS version 5.93, which was used to record all chromatograms.

The CNPE triterpenoid esters were separated using UHLP/SFC and detected using an ELSD due to the absence of chromophore groups.

Five octadecyl-bonded silica columns (75 cm of total length) used for the separation, including four Kinetex C18s (150 × 4.6 mm), 2.6 µm superficially porous particles from Phenomenex (Le Pecq, France) and one Accucore C18 (150 × 4.6 mm) with 2.6 µm superficially porous particles from Thermo-electron (les Ulis, France) were used in tandem based on previously published work [63,64,65]. Some of the conditions were modified from the above-mentioned method; the column oven temperature was set at 15 °C. Isocratic analyses were performed for 55 min with 80% SC-CO_2_ and 20% of the co-solvent composed of MeOH:ACN 75:25 (*v:v*). The total flow rate was 1.6 mL/min. The injection volume was set at 5 µL for all the samples. The back pressure regulator was set at 100 bar and heated at 60 °C to limit the effect of CO_2_ cold depressurization. The automated sampler was kept at 25 °C to avoid the precipitation of the extracts. The compounds were detected using an ELSD with a temperature of 40 °C, a filter of 4 s, a nitrogen pressure of 3 bars, and a gain set on wide.

To compare the yields of the CNPE, the peak areas of these two major terpenoid esters (faradiol-3-*O*-palmitate, faradiol-3-*O*-myristate) were evaluated (Figure 10).

#### 3.6.2. Unified Chromatography UC-DAD: Analysis of Polar Compounds in CNPE

To achieve a selective sequential SFE of non-polar compounds without the co-extraction of polar molecules, the CNPE needed to be evaluated for its content in targeted polar compounds. The analysis of polar compounds (narcissin) solubilized in the pure organic solvent (MeOH:DCM 1:1 *v:v*) in reverse phase UHPLC could lead to the precipitation of non-polar compounds in the column due to the presence of water in the mobile phase.

Therefore, to prevent this matter, all non-polar extracts were additionally analyzed using UC-DAD with an adapted gradient based on previously published work [66]. This was conducted using a Waters Corporation (Milford, Massachusetts, United States) ACQUITY Ultra Performance Convergence Chromatography™ (UPC^2®^) system equipped with a diode-array (ACQUITY PDA^®^). The analysis was carried out using a Torus DEA (100 × 2.1 mm; 1.7 μm) commercialized by Waters. The mobile phase was composed of CO_2_ and methanol acidified with 0.1% MSA, and the column oven was heated at 25 °C. Similarly, the automated sampler was also kept at 25 °C to avoid the precipitation of the extracts during the analysis sequence. This gradient is represented in Figure 11a. Due to the high inlet pressure generated at the end of the gradient by the high viscosity of the mobile phase that reached 100% of the organic solvent, reversed pressure and flow rate gradients were applied. The chromatograms were recorded using DAD in the 190–800 nm range. Visualization and peak integration were conducted at 354 nm Figure 11b. Since the column is of a polar nature (Diethylamine), only the polar compounds (narcissin and other flavonoids) interacted with the stationary phase and, therefore, were eluted later in the gradient with higher percentages of the modifier, while the non-polar compounds were not retained and eluted in dead time.

### 3.7. Calendula Polar Extracts (CPE): Chromatographic Analysis of Polyphenols

All polar extracts were analyzed with a Nexera-LC40 system of Shimadzu Corporation (Kyoto, Japan). This system was equipped with a photodiode array (PDA) detector (SPD-40), a solvent delivery unit (LC-40), an auto-sampler (SIL-40), a column oven (CTO-40), and a system controller SCL-40. All chromatograms were recorded on Lab-Solutions LC-UV 5.97 SP1 version (Shimadzu Corporation, Kyoto, Japan).

A Cortecs C18 (100 × 3.0 mm) column coupled to a Cortecs C18 VanGuard Cartridge (5 × 2.1 mm), both packed with 2.7 µm superficially porous particles, from Waters Corporation (Milford, MA, USA), were used for the analysis of all extracts. The column temperature was maintained at 30 °C. The injection volume was 5 µL, and the automated sampler was kept at 10 °C. The flow rate was maintained at 1 mL/min. The equilibration time between these two injections was 5 min.

The total time of each analysis was 11 min; the mobile phase consisted of a combination of H_2_O acidified with 0.1% of FA (solvent A) and ACN (solvent B). The percentage of solvent B varied as follows: 0–1 min: 10–25% B, 1–3 min: 25% B, 3–6 min: 90% B, 6.1–11 min: 10% B. The separation of CPE is presented in Figure 12.

The quantitative analysis of narcissin was performed by injecting a standard at 15 different concentrations from 0.001 mg/mL to 0.8 mg/mL. The calibration curve was obtained at 354 nm (*y* = 10^7^*x* + 48684, R^2^ = 0.9937), and the equation was used to estimate the concentrations from the peak area for narcissin (*y* = concentration; *x* = peak area). The total polar compound yield was estimated using the same equation as an equivalent to narcissin; this approximation was possible due to their similar molar absorption coefficient (ε) at 354 nm.

### 3.8. Experimental Design and Statistical Analysis

Ellistat software 6.4 2020/11 version (Poisy, France) was used for the experimental design and data analysis. A Box-Behnken design (BBD) with a response surface methodology (RSM) was chosen to establish the model and to determine the response pattern. It was used to optimize the supercritical fluid extraction of both polar and non-polar targeted compounds from *Calendula officinalis* L. flowers. For the optimization of CNPE, the extraction of three independent factors used in this study were temperature (*X*_1_), pressure (*X*_2_), and EtOH percentage as a polarity modifier (*X*_3_). As for CPE extraction, the three independent factors used were temperature (*X*′_1_), the total modifier percentage that consisted of an EtOH or EtOH:H_2_O mixture (*X*′_2_), and water percentage added as an additive to the modifier (*X*′_3_), while the pressure was kept constant at 15 MPa. The three levels that were used for both experimental designs were coded (+1) for the highest, (0) for the middle, and (−1) for the lowest levels (Table 1 and Table 3).

Both the regression model and the graphical analysis of the data were performed using the same software. The significance of the independent factors on the response was evaluated using an ANOVA analysis of variance. Significant factors were identified by a *p*-value of 0.05 or lower. This was performed using statistical tests such as Fisher’s test (F-value), and the classification of the significance of the model was performed following the contribution percentage of terms to the model. The fitness of the design model was evaluated using the correlation coefficient (R^2^), the adjusted correlation coefficient (adj R^2^), and the predicted R^2^ (prd R^2^). Two-dimensional heat map plots were used to illustrate the interaction between factors. The regression equation that represented the predictive model was solved to obtain the optimal extraction conditions.

The flow rate and plant mass were kept constant in all SFE experiments. Extraction durations were determined after kinetics of extraction at the central level experimental conditions (*X*_1_, *X*_2_, *X*_3_, *X*′_1_, *X*′_2_, and *X*′_3_ equal to 0). A total of 30 min was used for the CNPE design and 15 min for the CPE design.

For CNPE, the dependent response or output (*Y*) was the sum of the peak area for both major terpenoid esters (faradiol-3-*O*-palmitate, faradiol-3-*O*-myristate) found in marigold (mAU × min) for 30 min of extraction duration. As for CPE, the dependent response (*Y′*) was the yield of the target compound narcissin (mg/g) for 15 min of extraction duration. The experiments were randomized to maximize the effect of the variability in the response. Five replicates at the central level experimental conditions (*X*_1_ and *X*′_1_ = 0, *X*_2_ and *X*′_2_ = 0, and *X*′_3_ = 0) of the design were conducted to evaluate the experimental repeatability. The relative standard deviation (RSD) was estimated to confirm the reproducibility of the extraction model. The Dixon statistical test was applied to determine the presence of an outlier in the results with a 95% range.

Polynomial Equations (3) and (4) for CNPE and CPE, respectively, represented the relationship between the responses (*Y* and *Y*′) and the corresponding three independent variables (*X*_1_, *X*_2_, *X*_3_, *X*′_1_, *X*′_2_, and *X*′_3_). The equation terms were selected to represent the regression of the model while avoiding the overfitting of data. Consequently, for Equations (4) and (5), the identified terms were chosen to optimize the prediction of the model and, therefore, increase the predicted R^2^.
*Y* (mAU × min) = β_0_ + β_1_*X*_1_ + β_2_*X*_2_ + β_3_*X*_2_ + β_11_*X*_1_^2^ + β_22_*X*_2_^2^ + β_33_*X*_3_^2^ + β_13_*X*_1_*X*_3_(4)
*Y*′ (mg/g) = δ_0_ + δ_1_*X*′_1_ + δ_2_*X*′_2_ + δ_3_*X*′_2_ + δ_11_*X*′_1_^2^ + δ_22_*X*′_2_^2^ + δ_33_*X*′_3_^2^ + δ_23_*X*′_2_*X*′_3_(5)

With *β*_11_, *β*_22_, *β*_33_, *δ*_11_, *δ*_22_, and *δ*_33_ representing the quadratic coefficients, and *β*_13_ and *δ*_23_ representing the interaction coefficients between factors. The terms *X*_1_, *X*_2_, and *X*_3_ represent the following variables: temperature, pressure, and EtOH % used as a modifier, respectively. As for *X*′_1_, *X*′_2_, and *X*′_3_, they represent the following factors: temperature, modifier percentage, and water percentage in the modifier, respectively.

## 4. Conclusions

This study demonstrates the feasibility of sequential selective extraction of triterpendiol esters and polyphenols from *Calendula officinalis* L. flowers with supercritical fluids. Two experimental designs, adapted to the polarity of the targeted molecules, were used for the simple optimization of their extraction. First, the optimization of non-polar molecules allowed high-yield extracts to be recovered (around 90 mg/g), rich in major anti-edematous compounds (faradiol myristate and faradiol palmitate). This was achieved with the use of a high temperature of 80 °C combined with a pressure of 15 MPa and 15% EtOH in carbon dioxide, reaching a dried extract mass of 100.3 mg/g. The extraction of these non-polar compounds was scaled up to a pilot-scale extractor using 100 g of plant mass; the results of both analytical and pilot scales were similar when comparing solvent-to-feed ratios. Third, the optimization of polar compounds was conducted using a BBD, and a high concentration of polyphenols was reached in SFE at 40 °C, with 30% EtOH:H_2_O 80:20 *v:v*, for which 12.3 mg/g of polar compounds were recovered. This allowed for the sequential selective extraction of both non-polar and polar compounds of marigolds within a 60 min extraction with only two steps online. Finally, the SFE results were compared to UAE using comparable solvents and time conditions, showing that, for non-polar extraction, SFE was highly advantageous compared to UAE both in terms of extraction yields and the use of green solvents. For the polar UAE extraction, SFE had comparable results to UAE, showing the interest in using SFE for the extraction of polar molecules in addition to reducing water percentage in the final extract, favoring the stability of the extracts and reducing the energy cost of water evaporation.

## Figures and Tables

**Figure 1 molecules-28-07060-f001:**
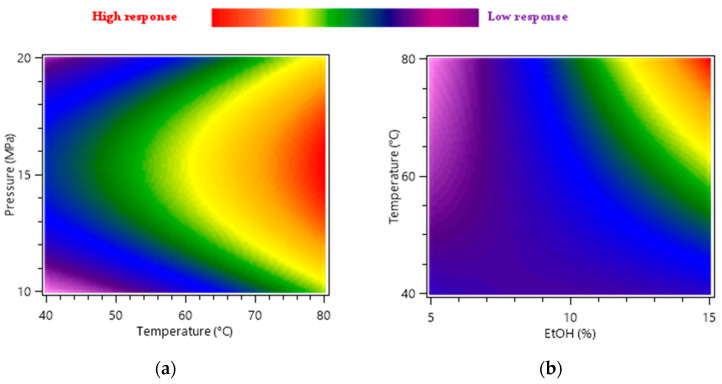
Response surface heat maps showing the effects of factor variation on the total faradiol esters (faradiol myristate and faradiol palmitate) and peak areas (mAU × min). (**a**) Interaction between pressure (*X*_1_) and temperature (*X*_2_), (**b**) Interaction between temperature (*X*_2_) and EtOH percentage (*X*_3_).

**Figure 2 molecules-28-07060-f002:**
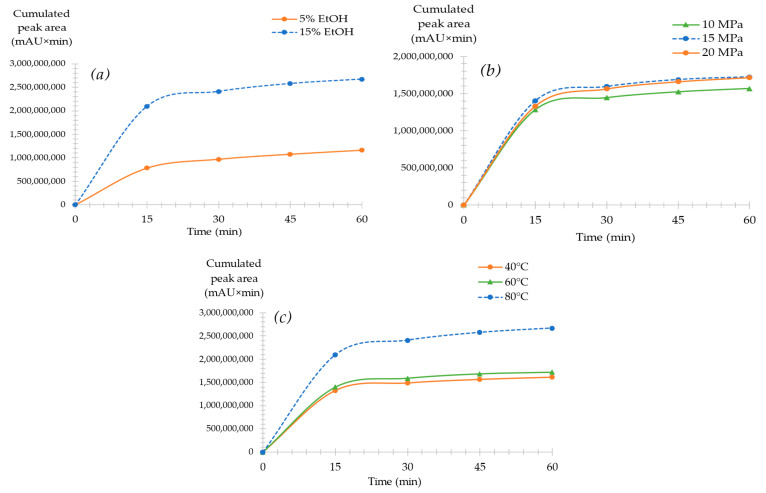
SFE kinetics of total fardiol esters (faradiol myristate and faradiol palmitate). (**a**) Variation in EtOH percentage (5 and 15%) at constant pressure (15 MPa) and temperature (80 °C). (**b**) Variation in pressure (10, 15, and 20 MPa) at a constant temperature of 60 °C and 15% EtOH. (**c**) Variation in temperature (40, 60, and 80 °C) at a constant pressure (15 MPa) and 15% EtOH.

**Figure 3 molecules-28-07060-f003:**
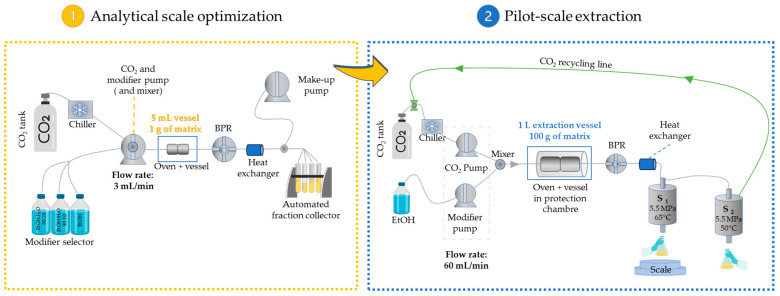
Schematic representation of the analytical scale extractor (AE) and pilot-scale extractor (PE) used in this study.

**Figure 4 molecules-28-07060-f004:**
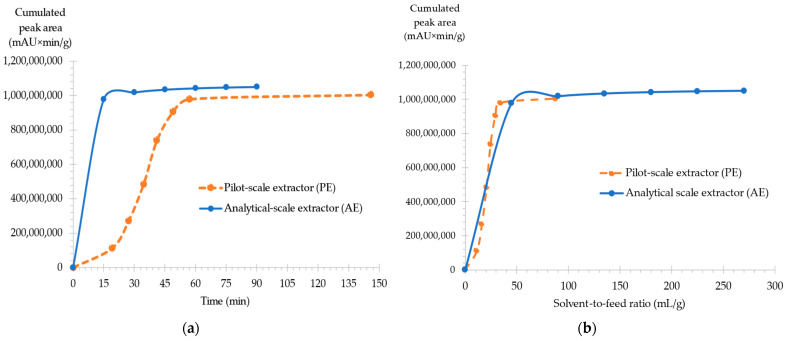
Comparison of cumulated peak areas (mAU × min/g) for major faradiol myristate and faradiol palmitate between analytical-scale extraction (AE) and pilot-scale extraction (PE) for CNPE. (**a**) Extraction kinetics of the total yield of faradiol (**b**) Variation in the total yield for the function of the solvent-to-feed ratio.

**Figure 5 molecules-28-07060-f005:**
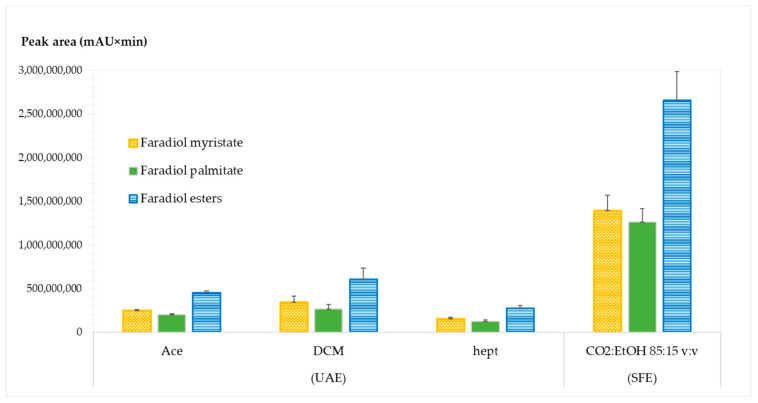
Comparison between supercritical fluid extraction (SFE) and ultrasound-assisted extraction (UAE) for CNPE faradiol esters (myristate and palmitate).

**Figure 6 molecules-28-07060-f006:**
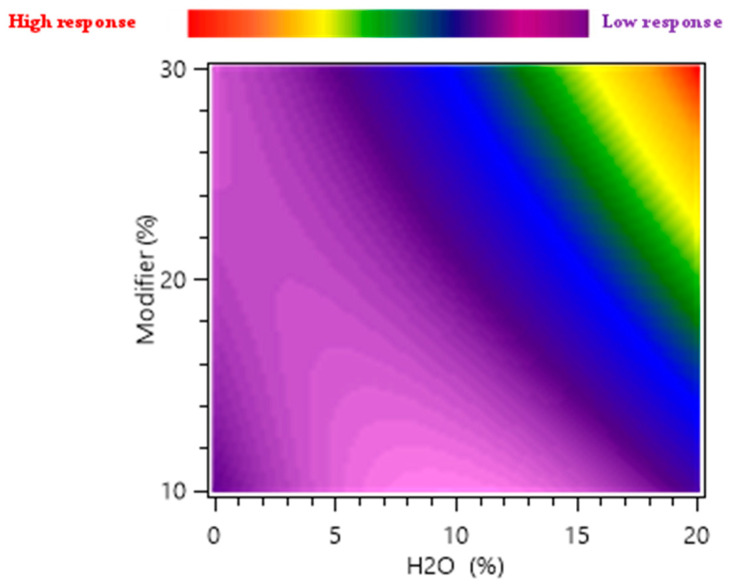
Response surface heat maps showing the effect of this factor’s variation on narcissin yield (mg/g). Variation in the total modifier percentage (*X*′_2_) and water percentage (*X*′_3_).

**Figure 7 molecules-28-07060-f007:**
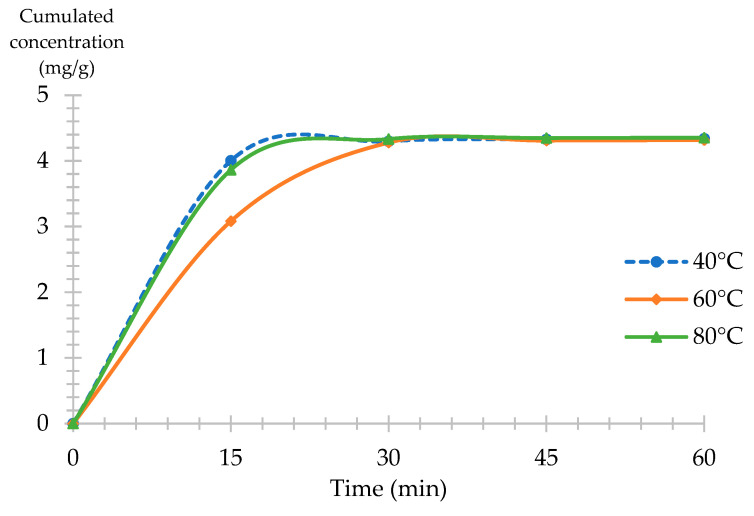
SFE kinetics of narcissin (mg/g) from CPE with a variation in temperature 40, 60 and 80 °C constant conditions: 30% EtOH:H_2_O 80:20 *v:v* as a modifier, 15 MPa, 3 mL/min.

**Figure 8 molecules-28-07060-f008:**
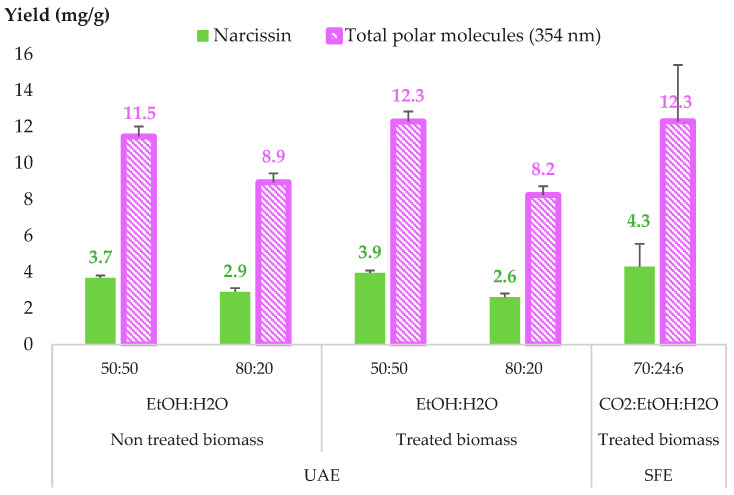
Comparison between supercritical fluid extraction (SFE) and ultrasound-assisted extraction (UAE) for polar compounds of CPE (total polar molecules represent all polar compounds quantified at 354 nm, including narcissin).

**Figure 9 molecules-28-07060-f009:**
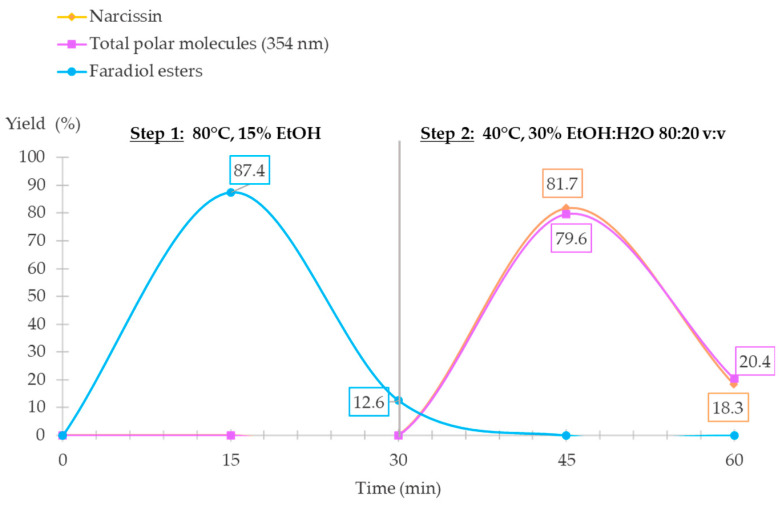
Extraction kinetics of the S^3^FE of non-polar and polar compounds from *Calendula officinalis* L. represented in a yield percentage for 60 min of extraction (total polar molecules represent all polar compounds quantified at 354 nm including narcissin).

**Figure 10 molecules-28-07060-f010:**
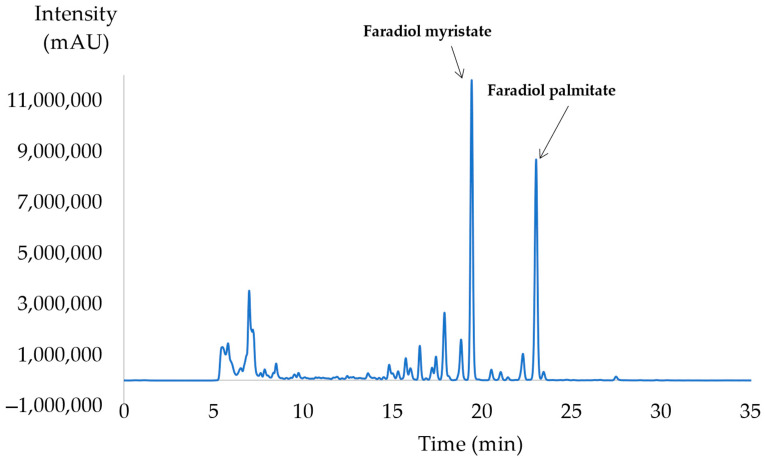
UHLP/SFC-ELSD analysis of the calendula non-polar extract (CNPE). Peak identification (1) faradiol-3-*O*-myristate (tr = 19.78 min), (2) fardiol-3-*O*-palmitate (tr = 23.46 min).

**Figure 11 molecules-28-07060-f011:**
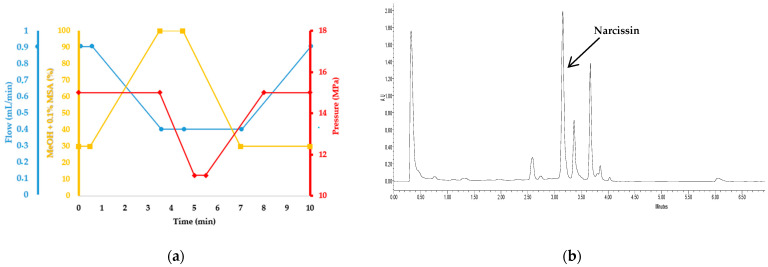
(**a**) UC-DAD gradient of polar compounds in CNPE. Blue: flow (mL/min), yellow: MeOH + 0.1% MSA (%) and red: pressure (MPa). (**b**) UC-DAD analysis of polar compounds in CNPE (354 nm)/SFE extract solubilized in MeOH:DCM 1:1 *v:v*. narcissin (tr = 3.15 min).

**Figure 12 molecules-28-07060-f012:**
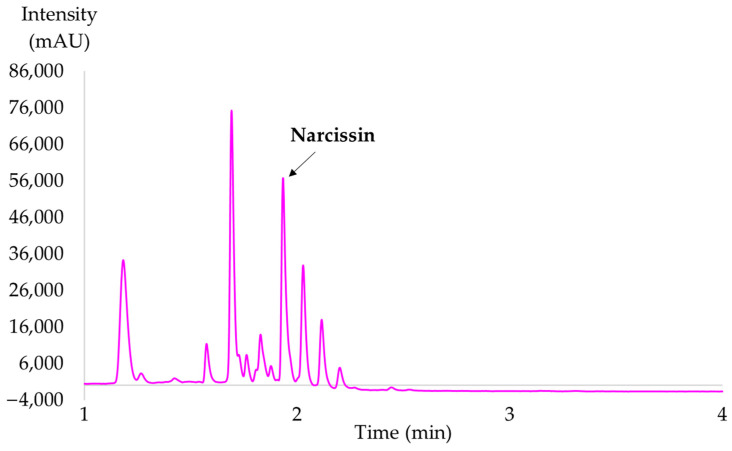
UHPLC-DAD analysis of polar extracts (354 nm); narcissin (tr = 1.93 min).

**Table 1 molecules-28-07060-t001:** Design matrix for the selective SFE design of the experiment (BBD) using CNPE with responses for non-polar compounds of interest (faradiol esters) and narcissin (UV response 354 nm UC-DAD).

Experiment N°	Coded Levels			Results (after 30 min of Extraction)				
	*X* _1_	*X* _2_	*X* _3_	Faradiol Myristate (mAU × min)	Faradiol Palmitate (mAU × min)	Both Faradiol Esters (*Y*) (mAU × min)	Narcissin (µv × s)	Dried Extract Yield (mg/g)
	Temperature (°C)	Pressure (MPa)	EtOH (%)					
1	40 (−1)	10 (−1)	10 (0)	3.11 × 10^8^	2.54 × 10^8^	5.64 × 10^8^	0.00	58.9
2	80 (+1)	10 (−1)	10 (0)	5.46 × 10^8^	4.65 × 10^8^	1.01 × 10^9^	4.08 × 10^5^	88.5
3	40 (−1)	20 (+1)	10 (0)	2.78 × 10^8^	2.31 × 10^8^	5.08 × 10^8^	0.00	62.5
4	80 (+1)	20 (+1)	10 (0)	5.50 × 10^8^	4.77 × 10^8^	1.03 × 10^9^	0.00	76.1
5	40 (−1)	15 (0)	5 (−1)	3.94 × 10^8^	3.27 × 10^8^	7.20 × 10^8^	0.00	56.5
6	80 (+1)	15 (0)	5 (−1)	1.99 × 10^8^	9.61 × 10^7^	2.95 × 10^8^	0.00	41.5
7	40 (−1)	15 (0)	15 (+1)	5.39 × 10^8^	4.66 × 10^8^	1.00 × 10^9^	3.23 × 10^5^	69.7
8	80 (+1)	15 (0)	15 (+1)	1.39 × 10^9^	1.26 × 10^9^	2.66 × 10^9^	0.00	92.3
9	60 (0)	10 (−1)	5 (−1)	1.30 × 10^8^	5.79 × 10^7^	1.88 × 10^8^	0.00	35.2
10	60 (0)	20 (+1)	5 (−1)	3.65 × 10^8^	3.13 × 10^8^	6.79 × 10^8^	0.00	55.5
11	60 (0)	10 (−1)	15 (+1)	6.81 × 10^8^	5.73 × 10^8^	1.25 × 10^9^	3.04 × 10^5^	85.4
12	60 (0)	20 (+1)	15 (+1)	6.50 × 10^8^	5.53 × 10^8^	1.20 × 10^9^	3.15 × 10^5^	102.6
13	60 (0)	15 (0)	10 (0)	5.77 × 10^8^	4.93 × 10^8^	1.07 × 10^9^	0.00	73.4
14	60 (0)	15 (0)	10 (0)	5.68 × 10^8^	4.68 × 10^8^	1.04 × 10^9^	0.00	70.7
15	60 (0)	15 (0)	10 (0)	5.82 × 10^8^	4.82 × 10^8^	1.06 × 10^9^	0.00	68.1
16	60 (0)	15 (0)	10 (0)	6.59 × 10^8^	5.49 × 10^8^	1.21 × 10^9^	0.00	71.5
17	60 (0)	15 (0)	10 (0)	4.59 × 10^8^	3.86 × 10^8^	8.45 × 10^8^	0.00	65.5

**Table 2 molecules-28-07060-t002:** ANOVA for response surface regression model and the total peak area of faradiol esters (faradiol myristate and faradiol palmitate) in selective CNPE.

Source	DF	SS	MS	F Value	*p*-Value	Contribution (%)	Conclusion
Regression model	9	4.3485 × 10 ^18^	6.2121 × 10^17^	17.5737	0.0001		Significant
*X*_1_—Temperature (°C)	1	5.9963 × 10^17^	9.8515 × 10^16^	2.7869	0.1294	3.75	Not significant
*X*_2_—Pressure (MPa)	1	1.9942 × 10^16^	4.0094 × 10^17^	11.3423	0.0083	15.3	Significant
*X*_3_—EtOH (%)	1	2.2417 × 10^18^	3.0974 × 10^17^	8.7624	0.0160	11.8	Significant
*X* _1_ ^2^	1	2.3401 × 10^15^	5.2683 × 10^15^	0.1490	0.7084	0.2	Not significant
*X* _2_ ^2^	1	3.7452 × 10^17^	3.8531 × 10^17^	10.9002	0.0092	14.7	Significant
*X* _3_ ^2^	1	3.3276 × 10^16^	3.3276 × 10^16^	0.9414	0.3573	1.27	Not significant
*X* _1_ *× X* _3_	1	1.0771 × 10^18^	1.0771 × 10^18^	30.4700	0.0004	41	Significant
Residuals	7	3.1814 × 10^17^	3.5349 × 10^16^			12.1	
Total	16	4.6666 × 10^18^					
	Degree of freedom	Sum of squares	Mean square				

**Table 3 molecules-28-07060-t003:** Design matrix for the selective SFE design of the experiment (BBD) and responses for polar compounds of interest from marigold flowers. Total polar molecules represent all polar compounds quantified at 354 nm, including narcissin.

Experiment N°	Coded Levels			Results (Concentration mg/g of Biomass after 15 min of Extraction at 354 nm)	
	*X*′_1_	*X*′_2_	*X*′_3_	Narcissin (Y′) mg/g of Biomass)	Total Polar Compounds (mg/g of Biomass)
	Temperature (°C)	Modifier (%)	H_2_O in Modifier (%)		
1	40 (−1)	10 (−1)	10 (0)	0.00	0.01
2	80 (+1)	10 (−1)	10 (0)	0.00	0.00
3	40 (−1)	30 (+1)	10 (0)	1.10	2.52
4	80 (+1)	30 (+1)	10 (0)	1.71	3.85
5	40 (−1)	20 (0)	0 (−1)	0.01	0.03
6	80 (+1)	20 (0)	0 (−1)	0.04	0.12
**7**	**40 (−1)**	**20 (0)**	**20 (+1)**	**3.31**	**9.18**
**8**	**80 (+1)**	**20 (0)**	**20 (+1)**	**2.82**	**8.14**
9	60 (0)	10 (−1)	0 (−1)	0.00	0.00
10	60 (0)	30 (+1)	0 (−1)	0.05	0.12
11	60 (0)	10 (−1)	20 (+1)	0.00	0.00
**12**	**60 (0)**	**30 (+1)**	**20 (+1)**	**5.30**	**14.64**
13	60 (0)	20 (0)	10 (0)	0.11	0.33
14	60 (0)	20 (0)	10 (0)	0.13	0.37
15	60 (0)	20 (0)	10 (0)	0.14	0.42
16	60 (0)	20 (0)	10 (0)	0.21	0.57

**Table 4 molecules-28-07060-t004:** ANOVA for response surface regression model of narcissin yield from selective CPE.

Source	DF	SS	MS	F Value	*p*-Value	Contribution (%)	Conclusion
Regression model	7	36.096	5.1566	34.3206	0.0000		Significant
*X*′_1_—Temperature (°C)	1	0.0028125	0.56408	3.7544	0.0887	3.02	Not significant
*X*′_2_—Modifier (%)	1	8.3232	0.22289	1.4835	0.2579	1.19	Not significant
*X*′_3_—H_2_O in modifier (%)	1	16.046	4.9439	32.9049	0.0004	26.5	Significant
*X*′_1_^2^	1	0.58141	0.58141	3.8697	0.0847	3.12	Not significant
*X*′_2_^2^	1	0.12076	0.12076	0.8037	0.3962	0.647	Not significant
*X*′_3_^2^	1	4.1311	4.1311	27.4951	0.0008	22.1	Significant
***X*′_2_ × *X*′_3_**	**1**	**6.8906**	**6.8906**	**45.8620**	**0.0001**	**36.9**	Significant
Residuals	8	1.202	0.15025			6.44	
Total	15	37.298					
	Degree of freedom	Sum of squares	Mean square				

## Data Availability

Not applicable.

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
