# Peer review of "Exploring the Sequential-Selective Supercritical Fluid Extraction (S3FE) of Flavonoids and Esterified Triterpenoids from Calendula officinalis L. Flowers"

_molecules, 2023, doi:10.3390/molecules28207060_

Round 1

Reviewer 1 Report

The work describes a study of a two-step consecutive extraction of biologically active compounds of different polarity from marigold flowers using supercritical fluid extraction (SFE) with ethanolic co-solvent (step 1) and with EtOH-H2O co-solvent (step 2). Full method development is presented along with extensive chromatographic analysis of extract fractions composition, extraction kinetic studies, process modeling as well as statistical scrutiny. The authors found that ethanolic co-solvent is necessary in SFE for the extraction of main marigold esterified triterpenes, faradiol myristate and faradiol palmitate. Optimization allowed creating a method far superior in extraction of non-polar compounds such as faradiol esters that ultrasonic assisted extraction (UAE) with either heptane, acetone or dichloromethane. Second stage of SFE employed EtOH-H2O mixed co-solvent which allowed extracting polar polyphenols, such as narcissin, in quantities comparable or even higher than UAE with water-ethanolic mixtures, and in less time.

The work is dedicated to an interesting topic, contains novel and significant results. Research is thoroughly planned, performed using comprehensive analytical and statistical methods, conclusive findings are scientifically significant and properly justified by abundant experimental data. Literary survey on marigold extraction research, including SFE, presented in the introduction is thorough and informative. The manuscript is well-written and presents the work without any notable lacunas. The paper should be accepted for publication in Molecules after addressing some minor issues listed below:

Remarks:

1.         Section 2.1.5, pilot-scale SFE of CNPE. I would suggest recalculating and re-illustrating extraction kinetic curves presented in Fig. 5a and 5b from time to solvent-to-feed ratio values (or, as authors called it, flow rate/plant mass ratio, line 322) on the abscissa axis. Direct comparison of extractions performed in different size instruments with different flowrate based on timescale is technically incorrect. Recalculating into solvent-to-feed ratio would allow performing a uniform comparison of analytical and pilot scales. As authors say in lines 321-325, the reason for the seeming difference in extraction kinetics is not some difference in matrix permeability, but simply the difference in flowrate per raw material mass unit. For instance, inflection points for AE curve in Figs. 5a and 5b are located at ~ 15 minutes, whereas for PE – at ~ 60 minutes, which seems much higher. But, 15 minutes in AE corresponds to solvent-to-feed ratio F = U*t/m(feed) = 3 mL/min*15 min: 1 g = 45 mL/g, whereas for PE F = 60 mL/min*60 min : 100 g = 36 mL/g, which is substantially smaller than for AE. It looks like PE is not slower, but actually faster that AE. That might lead to completely different conclusions on extraction kinetics on a pilot scale from the ones given at the moment in lines 311-312 and 332-334. If so, corresponding discussion should be corrected (lines 311-334).

2.         Section 2.1.6: authors compared SFE with CO2-EtOH to UAE with heptane, acetone and DCM solvents. Why not UAE with heptane-EtOH or heptane-iPrOH mixed solvent? It would probably be a closer solvent to CO2-EtOH, would make a more suitable comparison of extraction methods.

3.         Section 2.1.6, lines 342-343: why was solvent volume for UAE equalized to co-solvent volume in SFE? The author call it “equivalent conditions”, but it does not seem equivalent to exclude larger part of solvent volume from extraction process. Equivalent volume would be a volume of the whole CO2-EtOH mixture used during SFE. Please comment on that.

4.         Lines 418-420: As far as i can understand, it says here that, with 90/10 CO2-EtOH mixture, narcissin yield is 0.21 mg/g, whereas with 70/30 CO2-EtOH - it is 0 mg/g. Seems counter-intuitive, that increase in EtOH volume decreases narcissin yield, as well as that solvent as non-polar as 90/10 CO2-EtOH can extract any polyphenol glycoside at all. Could these 0.21 mg/g be some carryover from previous extractions with more polar solvents?

5.         Lines 545-546: the phrase is unclear and should be rewritten. What does “every 50 g of extract” mean? If 6 fractions were collected, extraction time was 57 minutes and total flowrate was 60 ml/min – then, fractions were collected, roughly, every 9 minutes, correct?

6.         Fig. 9: please fix the signatures of two main leaks on the chromatograms. Arrows and texts pointing at two main peaks are shifted from correct positions.

Notices mistypings:

o   Lines 12, 109, 295, 441, 453, 483, 553, 663, 710: calengula officinalis is written with small “c” letter, it should be capital.

o   Line 151: “; it was demonstrated that”

o   Line 70: Sentence structure is unclear. Why “Nevertheless”? As opposed to what?

o   Line 71: “fatty acid content … has been …”

o   Line 553: “powdered calendula officinalis L. powder” – two times “powder”.

o   Line 674: “This was done using statistical tests such as Fisher's test

Reviewer 2 Report

The manuscript “Exploring the Sequential-Selective Supercritical Fluid Extraction (S3FE) of flavonoids and esterified triterpenoids from Calendula officinalis L. flowers” summarizes the supercritical fluid extraction (SFE) technology of non-polar and polar compounds from f calendula officinalis L. the authors evaluated the factors effects for selective extraction: for non-polar compounds the factors were pressure, temperature and EtOH percentage and for the polar compounds the factors were temperature, total modifier percentage and H2O added in modifier as an additive. In my opinion, the article is not complete enough to be published in a high impact journal like Molecules.

Summary: The summary is incomplete. The authors do not conclude about the different extraction methods and do not report the yield values obtained.

Introduction: In the introduction, the advantages of SFE are clear. However, on page 1, lines 32 to 35, the authors mention that other extraction methods have some limitations such as "low recovery and extraction selectivity". Depending on the method and solvent used, this statement may not be true.

Why do the authors use the term modifier and not co-solvent?

More references on biological activities could be included. It is requested to add reference material about biological applications of molecules, to bring greater scientific relevance to the introduction. Yield and other physicochemical characteristics of molecules are highly dependent on extraction and isolation techniques and on the source. In addition, it is important to present the extraction methods, which parameters influence them and how they influence the structure of compounds.

The authors claim that SFE, compared to other extraction methods, generally has the advantages of high extraction efficiency, saving time, and low energy/solvent consumption. SFE is an expensive technique and its choice needs to be justified.

Results: Table 2: In my opinion, authors should only use “significant” or “non-significant”.

In 2.1.3. Optimal selective extraction of faradiol esters: what do the authors mean by selective extraction? How is selectivity calculated?

On page 8, line 267: “..demonstrating that a high percentage of modifier at 80°C is needed”. Could a higher percentage of ethanol be tested?

In Figure 2c, why does temperature cause an increase in extraction? This temperature could cause degradation or modification in molecules structure. The authors could address in more detail the structure of the molecule and how its properties can be affected by extraction methods, such as interaction with solvents or temperature, for example.

Table 2. Pressure was the factor in had the highest contribution to extraction. Since temperature was not significant, it is better to use lower temperature to favor lower energy consumption.

Pg 10, line 323 The authors could fix the solvent/plant ratio to study the scale-up.

Table 4:  The factor H2O percentage in modifier was significant. What is the effect of water on extraction yield. What is the optimal percentage to use?

2.3. Validation of SFE , line 489: “The first step used 15% EtOH at 80°C ….”. Could 40°C be used in the first stage?

Methology: The article is confusing, and the authors could add a schematic diagram explaining the Steps.

Conclusion:  The authors should compare the yield obtained with those already reported in the literature using other extraction methods.

They could also compare the antioxidant activity of the compounds obtained by the two methods and propose an application for the extracted molecule.

English language should be improved.

Author Response

please see the attachement

Reviewer 3 Report

This study used the sequential SFE to extract both non-polar and polar compounds. Two BBD studies were adopted for screening the optimal conditions. I recommend publishing this manuscript after revision. The following are my suggestions.

1.      In Table 1, the units for the content of extraction targets are mAU x min or mV x sec. Can this unit be converted to concentration or mg/g to show its physical meaning?

2.      In both BBD studies, in addition to R2, I recommend adding the results of the lack of fit test to validate the proposed equations (equations 1 and 2)

3.      In Tables 1 and 3, I recommend adding calculated response (Y) values using proposed equations (equations 1 and 2) to compare with the experimental data.

4.      In Table 2, terms X1X2 and X2X3; in Table 4, terms X1X2 and X1X3 were excluded for developing the equations for response. Please give an illustration from statistics.

5.      Please discuss the mechanism of the significant cross-interaction effect of X1X3 in Table 2 and X2X3 in Table 4.

6.      Since the properties of raw materials used in SFE are crucial. In section 3.1, please also mention the properties of dried flowers, such as the particle size characteristics and water content.

7.      Please check the label of the two main components in Fig. 9.

8.      There are many methods for process optimization. Why BBD was selected in this study?

Reviewer 4 Report

This manuscript deals with supercritical fluid extraction of flavonoids and esterified triterpenoids from Calendula officinalis L. flowers in the analytical and pilot-scale extractors. The dada gathered are eough to be published in this Journal, but some points should be addressed.

-The major processing parameter in SFE is pressure, and the minors are temperature and cosolvent. In this research, the pressure is low(10, 15, 20 MPa) and the solvent (ethanol) is too high (15%). Needs to try to reduce the usage of organic solvents as can as possible.

-Remove "Sequential-Selective" in the title of this manuscript.

Only just collecting extracts sequentially. Nothing more than that.

-The authors should express the concentration of each flavonoids in mg/g instead of mAU×min in all tables.

-Add schematic diagrams of the analytical and pilot-scale SFEs.

-For flavonoids extraction by organic solvents, common solvents are methanol and acetone. Explain the reason using heptane and dichloromethane.

Round 2

Reviewer 2 Report

In my opinion, the revised version of the manuscript was sufficiently improved to warrant publication in Molecules.

Reviewer 4 Report

..